# Analysis of Outbreak and Global Impacts of the COVID-19

**DOI:** 10.3390/healthcare8020148

**Published:** 2020-05-29

**Authors:** Ishaani Priyadarshini, Pinaki Mohanty, Raghvendra Kumar, Le Hoang Son, Hoang Thi Minh Chau, Viet-Ha Nhu, Phuong Thao Thi Ngo, Dieu Tien Bui

**Affiliations:** 1Department of Electrical and Computer Science, University of Delaware, Newark, DE 19716, USA; ishaani@udel.edu; 2Department of Computer Science, Purdue University, 610 Purdue Mall, West Lafayette, IN 47907, USA; mohantypinakiranjan@gmail.com; 3Computer Science and Engineering Department, GIET University, Gunupur, Odisha 765022, India; raghvendraagrawal7@gmail.com; 4VNU Information Technology Institute, Vietnam National University, Hanoi 010000, Vietnam; sonlh@vnu.edu.vn; 5Faculty of Information Technology, University of Economic and Technical Industries, Hanoi 010000, Vietnam; hgchau85@gmail.com; 6Geographic Information Science Research Group, Ton Duc Thang University, Ho Chi Minh City 700000, Vietnam; 7Faculty of Environment and Labour Safety, Ton Duc Thang University, Ho Chi Minh City 700000, Vietnam; 8Institute of Research and Development, Duy Tan University, Da Nang 550000, Vietnam; ngotphuongthao5@duytan.edu.vn; 9Department of Business and IT, University of South-Eastern Norway, Gullbringvegen 36, N-3800 BøiTelemark, Norway; Dieu.T.Bui@usn.no

**Keywords:** corona virus, global impacts, global pandemic, COVID-19, severe acute respiratory syndrome (SARS)

## Abstract

Corona viruses are a large family of viruses that are not only restricted to causing illness in humans but also affect animals such as camels, cattle, cats, and bats, thus affecting a large group of living species. The outbreak of Corona virus in late December 2019 (also known as COVID-19) raised major concerns when the outbreak started getting tremendous. While the first case was discovered in Wuhan, China, it did not take long for the disease to travel across the globe and infect every continent (except Antarctica), killing thousands of people. Since it has become a global concern, different countries have been working toward the treatment and generation of vaccine, leading to different speculations. While some argue that the vaccine may only be a few weeks away, others believe that it may take some time to create the vaccine. Given the increasing number of deaths, the COVID-19 has caused havoc worldwide and is a matter of serious concern. Thus, there is a need to study how the disease has been propagating across continents by numbers as well as by regions. This study incorporates a detailed description of how the COVID-19 outbreak started in China and managed to spread across the globe rapidly. We take into account the COVID-19 outbreak cases (confirmed, recovered, death) in order to make some observations regarding the pandemic. Given the detailed description of the outbreak, this study would be beneficial to certain industries that may be affected by the outbreak in order to take timely precautionary measures in the future. Further, the study lists some industries that have witnessed the impact of the COVID-19 outbreak on a global scale.

## 1. Introduction

A pandemic may be defined as a global outbreak of a disease. In the past, several diseases such as Ebola virus disease, plague, severe acute respiratory syndrome (SARS), etc. have been the cause of a global epidemic and resulted in millions of deaths. The latest disease that contributes to the global pandemic is the corona virus or the COVID-19 [1]. Corona viruses are a large family of viruses that are responsible for causing illness that can range from the common cold to more severe diseases such as Middle East respiratory syndrome and severe acute respiratory syndrome [2]. The corona virus disease 2019 (COVID-19) is called the novel corona virus (2019-nCoV) since it is a new type and has not been identified before [3]. Detailed investigations point out that SARS-CoV was transmitted from civet cats to humans and MERS-CoV from dromedary camels to humans [4,5,6,7,8,9]. Common signs of infection include respiratory symptoms, fever, cough, shortness of breath, and breathing difficulties [10]. In more severe cases, the infection can cause pneumonia, severe acute respiratory syndrome, kidney failure, and even death. 

Corona viruses are zoonotic, which means that they are transmitted between animals and people [11]. The first was identified in Wuhan [4]. The infectious disease is not restricted to Wuhan city anymore but has spread across the globe, making an impact in almost all the continents, resulting in thousands of deaths [12]. Further, it has impacted several industries like the global supply chain, restaurant business, tourism, e-commerce, etc. While several researchers assert the origin of the corona virus to bats [13], it has also been attributed to the animal and seafood market [14,15]. Since the origin is not known, understanding the infectious disease is a challenge. What is more difficult is developing a vaccine for the same [16]. This has led to an increase in the number of deaths over the last couple of months, and the death toll is expected to rise even more. Thus, analyzing the global pandemic is a must. While the outbreak is only a few months old, the gravity of the situation has led to several research works on the same, which we will discuss in this section.

The preliminary identification of potential vaccine targets has been proposed for the novel corona virus based on severe acute respiratory syndrome-associated corona virus (SARS-CoV) immunological studies [17]. Although their findings may help guide experimental efforts toward the development of vaccines against the novel corona virus, theoretically, since the same has not been tested, there might be a question of feasibility for the research. A study [18] on three corona virus diseases in the last two decades, i.e., SARS, MERS, and COVID-19 highlighted that the case fatality ratio for SARS-CoV was calculated to be 9.5%, whereas for MERS-CoV (2012), the case fatality ratio was 35%. The case fatality ratio of COVID-19 (2019) is 2–3%. 

It has been found that the related research works [19,20,21,22,23,24,25,26,27,28,29,30,31] were based on summarized case studies and reports. In this paper, we perform an in-depth analysis of the COVID-19 outbreak and intensity based on the cases confirmed, cases recovered, and cases that resulted in death all over the globe. The analysis has been supported by means of data visualization (graphs and heat maps). Secondly, many of the findings are flawed due to a lack of data. Since our findings are based on a dataset released by John Hopkins University, the data might not be flawed for our research. Thirdly, several research papers limit their study to only China. In our research, we exhibit our findings on a global scale. Fourthly, many of the research papers deal with either the number of deaths or the number of confirmed cases, but not both. In our research paper, we take into account the number of confirmed cases, the number of deaths, and also the number of recovered cases. Fifthly, the findings of the past research work are based on limited data. As the number of cases increases for corona virus, the dataset is extended. In our research, we use the latest extended dataset (22 January 2020 to 4 April 2020). Lastly, a variety of visual representations have been provided such as line graphs, heat maps, and bar graphs so as to validate the study. Several industries such as travel and tourism, film industry, aviation industry, etc., which have been affected due to the outbreak, will be discussed in the paper.

In this paper, we analyze COVID-19 outbreak behavior over the last couple of months by considering the outbreak cases (confirmed, recovered, deaths). A visualization of how the outbreak has spread and its impact over the globe has been detailed in this article. This paper also discusses the COVID-19 outbreak in China as well as for the rest of the world. 

## 2. Material and Methods

### 2.1. Material

The datasets were taken from the Humanitarian Data Exchange [32] and incorporate novel corona virus (COVID-19) epidemiological data since 22 January 2020. We analyzed the data from 22 January 2020 to 4 April 2020. The data are compiled by the Johns Hopkins University Center for Systems Science and Engineering (JHU CCSE). Various sources of this data have been taken from the World Health Organization (WHO), DXY.cn. Pneumonia. 2020, BNO News, National Health Commission of the People’s Republic of China (NHC), China Center for Disease Control and Prevention (CCDC), Hong Kong Department of Health, Macau Government, Taiwan CDC, U.S. CDC, Government of Canada, Australia Government Department of Health, European Center For Disease Prevention and Control (ECDC), and Ministry of Health Singapore (MOH). JSU CCSE also maintains this data on the 2019 novel corona virus COVID-19 (2019-nCoV) data repository on GitHub. Three datasets have been made available, which include fields Province/State, Country/Region, Last Update, Confirmed, Suspected, Recovered, Deaths. For our research, we used four different types of data:Cases confirmed: The total number of corona virus cases that have been confirmed worldwide.Cases that resulted in death: The total number of corona virus cases that resulted in the death of the patients worldwide.Cases recovered: The total number of corona virus cases that resulted in the recovery of the patients, worldwide.Locations (Latitude and Longitude): Number of cases confirmed, recovered, and resulting in death based on locations worldwide. 

These data will be used for visually representing how the COVID-19 pandemic has spread worldwide. There are several ways to analyze the COVID-19 outbreak. For our research, we relied on the following:Graph Plots: We depict the number of cases confirmed, recovered, and resulting in death over a span of almost two months (22 January 2020–15 March 2020). The graph plots will be beneficial in observing the latest trends of the COVID-19 outbreak.World Heat Maps: Based on the number of cases as per the location specified in the dataset a world heat map may be generated to visually show the spread of COVID-19. The heat map depicts the density of the pandemic in different locations of the globe.

In the later part of the Results and Findings sections, we present some graphs that have been plotted using data from Statista, which is an online portal for statistics that incorporates data collected by the market and other research institutes. It includes quantitative data related to media, business, finance, politics, and a wide variety of other areas of interest or markets. While the datasets have been taken from Statista, the data have been accumulated from reliable sources such as the World Health Organization (WHO), OpenTable, The Hollywood Reporter, Official Airline Guide (OAG), RaidióTeilifísÉireann (RTE) Media services, etc. The graphs manifest how the pandemic has affected some industries on the global scale.
The revenue figures across the globe for the travel and tourism industry were taken from cruises, hotels, package holidays, and vacation rentals. The estimations were based on IATA’s (International Air Transport Association) forecast on the flight industry.The decline in the number of seated diners is based on a survey performed by OpenTable (online restaurant-reservation service company) and the data pertain to online reservations, phone reservations, and walk-ins. OpenTable is known to be active in more than 80 countries. Australia, Canada, Germany, Ireland, Italy, Japan, Mexico, The Netherlands, Spain, UK, and the United States are some of the countries where OpenTable is active.The estimated revenue loss of the film industry worldwide is based on a survey performed by The Hollywood Reporter (American digital and print magazine and website).The flight frequency of global airlines is based on a survey performed by the Official Airline Guide (OAG) schedule analyzer. OAG is the world’s leading provider of digital flight information, intelligence, and analytics for airports and airlines.The potential cost of the Tokyo 2020 Olympics is based on a survey performed by RTE (RaidióTeilifísÉireann), which is a national public service media of Ireland.The potential loss of revenue for Formula One is based on a survey performed by Forbes, which is a leading global media company.The estimated loss of revenue for National Hockey League (NHL) teams is based on a survey performed by The Athletic, which is a sports journalism website.

### 2.2. Methods

As we mentioned in the dataset section, our dataset includes the fields Province/State, Country/Region, Last Update, Confirmed, Suspected, Recovered, Deaths. Based on these fields, we visually represented the data in several forms ranging from tree maps to world heat maps. The graphs depict the number of cases confirmed with respect to time, number of cases recovered with respect to time, and number of deaths that occurred with respect to time. We considered the number of days as time here (22 January 2020 to 4 April 2020). The world heat map is a visualization of the severity (by number) of the cases (confirmed, recovered, and death) specific to the frequency of occurrences of locations mentioned in the dataset. Several steps were performed in order to generate the graphs as follows:Data Collection: As mentioned in the previous section, the data were collected from the repository maintained by John Hopkins University. The datasets include the fields Province/State, Country/Region, Last Update, Confirmed, Suspected, Recovered, Deaths, etc. (Table 1).Data Preprocessing and Cleaning: The data were compiled to provide information about Province/State, Country/Region, Latitude, Longitude, Date, Confirmed, Death, and Recovered.Deriving Additional Columns: Once the above data are preprocessed and cleaned, we obtained an additional data field, i.e., Number of Active cases. The new condensed data incorporate information about the date, confirmed cases, deaths, recovered cases, and active cases (Table 1).Additional Derived Data: Similarly, additional data such as top ten countries that have been affected by the COVID-19 outbreak or the number of countries where there are no death cases can be derived accordingly.

## 3. Results and Discussion 

Based on the new dataset and the derived datasets, we are able to generate some graphs and study the behavior of the COVID-19 outbreak across the globe. Since it originated in China, we compare how COVID-19 has affected China as well as the rest of the world.

### 3.1. Results

In this section, we present some plots, graphs, and heat maps depicting the outbreak of the COVID-19 pandemic. Data visualization through these plots, graphs, and heat maps may be used to communicate information pertaining to the COVID-19 spread clearly and efficiently. 

#### 3.1.1. COVID-19 Spread across the Globe

Figure 1 is a representation of the COVID-19 outbreak across the globe with three cases (confirmed, recovered, and deaths). The number of confirmed cases has increased ever since the virus was identified. While the recovery rate has shown significant increase, mortality (number of deaths) has increased the least over the last two months. 

#### 3.1.2. COVID-19 Spread: Inside China (Cumulative Number of Cases)

Figure 2 is a depiction of all cases within China depicted separately. As we can see, compared to the number of confirmed cases, a large number of patients recovered. The number of deaths is relatively less. There may be some who are active and are receiving treatment.

#### 3.1.3. COVID-19 Spread: Outside China (Cumulative Number of Cases)

Figure 3 is a depiction of all cases outside China depicted separately. As we can see, compared to the number of confirmed cases, a meagre portion of cases recovered while some cases resulted in death. There may be some patients who are active and are receiving treatment.

#### 3.1.4. COVID-19 Spread across China 

Figure 4 is a depiction of all the three cases (confirmed, recovered, and deaths) in China. While the virus was first identified in China, it has impacted different countries all over the world. As we can see, initially the number of cases increased significantly, after which it started decreasing. Over the last one month, the numbers of recovered cases have been dominant while the numbers of confirmed cases and death cases have reduced.

#### 3.1.5. COVID-19 Spread Outside China 

Figure 5 is a depiction of all the three cases (confirmed, recovered, and deaths) outside China. Initially, the number of cases increased significantly, after which it started decreasing slightly. The number of recovered cases has also increased slightly. Although the number of deaths reported is less as compared to the number of confirmed cases, on a global scale, it is still consequential.

#### 3.1.6. COVID-19 Spread across the Globe: Number of Countries Affected

Figure 6 depicts the number of countries that have been affected due to the outbreak. As of 15 March 2019, it has been observed that a total of 146 countries have been affected by the outbreak. COVID-19 has spread across all continents (except Antarctica) affecting 181 countries by 4 April 2020.

#### 3.1.7. COVID-19 Spread across the Globe: Number of Countries Affected by Cases Confirmed, Cases Recovered, and Death Cases 

Figure 7a depicts the spread of COVID-19 across countries all over the world. As per the scale, as the intensity of the number of confirmed cases increases, regions get darker in the map. It may be asserted that countries such as China, the United States, and parts of Europe have many confirmed cases with respect to countries in South America. The difference in intensity depicts the number of confirmed cases. As the colors get darker, more and more confirmed cases are reported. Figure 7b depicts the number of recovered cases across the world. On a scale of 40, it may be estimated that countries such as China, United States, parts of South America have many recovered cases with respect to countries in Africa and South America. The difference in intensity depicts the number of recovered cases. As the colors get darker, more and more recovered cases are reported. Figure 7c depicts the number of death cases across the world. On a scale of 40, it may be estimated that countries such as China, United States, parts of Europe have reported more deaths with respect to countries such as South Africa and South America. The difference in intensity depicts the number of recovered cases. As the colors get darker, more and more death cases are reported.

#### 3.1.8. COVID-19 Spread across the Globe: Cases Confirmed, Cases That Resulted in Death, Cases Recovered, and Cases Active Globally

Figure 8a is a bar graph representation of the number of confirmed cases across the world. The United States has the maximum number of confirmed cases (greater than 275,000), followed by Italy (greater than 119,800), followed by Spain (greater than 119,100). Figure 8b is a bar graph representation of the number of death cases across the world. Italy has the maximum number of death cases (greater than 14,000), followed by Spain (greater than 11,000), followed by the United States (greater than 7000). Figure 8c is a bar graph representation of the number of recovered cases across the world. China has the maximum number of recovered cases (greater than 76,000), followed by Spain (greater than 30,000), followed by Germany (greater than 24,000). Figure 8d is a bar graph representation of the number of active cases across the world. The United States has the maximum number of active cases (greater than 250,000), followed by Italy (greater than 85,000), followed by Spain (greater than 77,000). 

#### 3.1.9. COVID-19 Spread across the Globe: Number of Deaths per 100 Confirmed Cases

Figure 9 is a bar graph representation of the number of deaths per 100 confirmed cases. Italy has the highest mortality rate (12.25), followed by the San Marino (12.24), followed by France (10). While the disease was identified for the first time in China, the number of deaths per 100 confirmed cases seems to have more impact in other countries.

### 3.2. Findings

Based on the graphs plotted and the heat maps generated, several observations can be made. The observations are as follows:The number of cases confirmed for COVID-19 globally is higher than the number of cases recovered, followed by the number of cases that resulted in deaths.The graphs depict how the number of cases confirmed, recovered, and resulted in death have only increased (it has not been constant).The percentage of cases where patients recovered with respect to the number of cases confirmed is calculated to be approximately 20.40%.According to [18], the case fatality ratio of COVID-19 is 2–3%. According to our analysis based on recent data, the number of cases where patients died with respect to the number of cases confirmed is found out to be 58,787/1,095,915 = 0.0536, which is 5.36%. The variation depicts how the pandemic outbreak has affected a large number of people in a short period of time.The percentage of cases where patients are active with respect to the number of cases confirmed is calculated to be 74.23%.The world heat maps generated depict how the cases (confirmed, recovered, death) have spread globally. Darker colors represent higher intensity of the cases reported.While COVID-19 was first identified in Wuhan district of China, it has spread across the globe. The graphical representations justify by numbers and regions how the pandemic has an impact over the world population.

### 3.3. Global Impacts of the COVID-19 Outbreak

The corona virus was detected on 31st December 2019 in Wuhan City, Hubei Province, China. In the last couple of months, the number of cases has astonishingly increased to 4000 cases in Wuhan City alone (uncertainty range: 1000–9700) [26]. The pandemic was not confined to Wuhan city or Hubei Province or China, but managed to spread globally, such that every continent (except Antarctica) has COVID-19 cases confirmed. The number of confirmed cases rose to 78,000 [33] and now more than a million people have been affected by the outbreak. The COVID-19 outbreak not only resulted in more than three thousand deaths [34], but also affected several industries. Since the COVID-19 outbreak is highly attributed to infected passengers traveling internationally, the travel industry has been one of the most affected industries. This has led to travel restrictions for limiting the spread [35]. Multiple countries fear the risk of transportation [36] of infected patients and perform screenings at airports [37]. With globalization interacting and interconnecting people, businesses, and nations, many industries become interconnected and interdependent. Therefore, an outbreak affecting a particular industry may also have a severe impact on other connected industries. In this section we have considered some industries such as the tourism industry, restaurants and leisure (industry), entertainment industry, travel industry, and sports industry. The outbreak may have a significant effect on each of these industries. The study may also be useful in anticipating what other industries that are connected to the industries taken in consideration may be impacted for future research. As mentioned in Section 3.1, the data have been taken directly from Statista, an online portal for statistics that incorporates data collected by market and research institutes. Since for many industries we are comparing revenues, we have used bar charts (horizontal and vertical) to depict the same. The bar chart comparison based on statistics depicts the predicted revenue and the actual revenue for most industries. In this section, we discuss a few industries that were affected due to the deadly outbreak as follows.

#### 3.3.1. Tourism Industry

With the outbreak of the corona virus, international tourism may slump particularly in China, Italy, and South Korea [38]. Governments could get busy quarantining towns and cities and shutting down transportation. Travel warnings may be issued for anyone considering visiting countries that have several COVID-19 cases reported, which could put travelers on high alert, and many may not prefer to travel. Since corona virus has spread across the globe, the revenue of tourist destinations around the world may be hit hard. The corona virus may cost the U.S. travel industry alone billions over the next few years [39]. Figure 10 depicts how the outbreak has an impact over global revenue for the travel and tourism industry.

Figure 10 depicts three continents that have been affected the most due to the COVID-19 outbreak: North America, Europe, and China. As is evident from the graph above, there is a decline in global revenue for the travel and tourism industry due to the COVID-19 outbreak. A bar graph has been used to show the comparison between the revenues for the years 2019 and 2020. We observe that the decline is 9.41% percent for North America; and for Europe, the decline is 16.18%. Asia witnessed the maximum decline in revenue, which is estimated to be 27.08%, and therefore has suffered most in the case of the tourism industry. The estimates are based on IATA’s forecast regarding the overall effect of COVID 19 on the flight industry. As the demand for flights started decreasing, the number of leisure trips also declined leading to a loss in revenue. 

#### 3.3.2. Restaurants and Leisure

The restaurant sector in China may witness a significant fall due to the corona virus outbreak. The incident coincided with this year’s Spring Festival. The city of Wuhan was sealed off and quarantined on January 23, following restaurants all over the country closing their shutters. Since the outbreak has affected people on a global scale, it has led to people socializing less, thus leading to many restaurants being shut down or providing limited services. Figure 11 depicts the impact of COVID-19 spread over restaurants on a global scale.

The corona virus (COVID-19) pandemic has caused significant damage to the global restaurant industry. Due to measures of social distancing and general caution in public places, consumers have been dining out less and less. This has also led to fewer staff in the restaurants, thus jeopardizing the jobs of several people working in the restaurants. According to Statista and Open Table, the decline of seated diners in restaurants worldwide lowered to 56% on 16 March 2020 and was all the way down to 100% by the end of March [40].

#### 3.3.3. Entertainment Industry 

With the outbreak of corona virus, the lucrative Chinese film industry could also suffer. As movie theaters across the country are closed, major releases may eventually get delayed. It may have an impact on other entertainment industries too, and the impact may be on a global level. The entertainment industry could see lowered attendance at film festivals and disruptions in film distribution to delayed or canceled movie releases. Studios, filmmakers, and theater owners may also be affected financially. Figure 12 depicts the impact of COVID-19 spread over the entertainment industry on a global scale.

The global film industry has suffered a revenue loss of seven billion U.S. dollars as of the middle of March 2020 worldwide due to the COVID-19 outbreak. With theaters closing, movie premieres being postponed, screenings canceled, and box offices closed, it is estimated that the Film Industry may lose ten billion dollars in revenue by the end of May 2020 [41].

#### 3.3.4. Travel Industry 

Being one of the largest industries in the world, the travel industry has already taken a huge hit due to travel restrictions and canceled trips for both business and pleasure. Airlines have cut hundreds of thousands of flights because of corona virus travel restrictions. Carriers also are waiving change and cancellation fees for some routes because of the illness [42]. The impact of COVID-19 over airlines industry can be seen in the following Figure 13.

The impact of the COVID-19 outbreak has witnessed changes in the frequency of flights all over the world. Since multiple flights are taken into account, we have used the line chart to depict the change in flight frequency of global airlines (Figure 13). As is seen in the graph above, the week starting from 9 March 2020, the number of scheduled flights worldwide was down by 10.1%. The Chinese aviation reached a peak in the week starting 17 February 2020, with flight numbers down by 70.8%. Toward the end of March 2020, Italian aviation went down by more than 80%.

#### 3.3.5. Sports Industry 

The COVID-19 outbreak is likely to have a significant impact on the sports industry. Since the sports industry witnesses a lot of gatherings over many games and competitions all over the world, there is a probability that the outbreak may infect a large number of people in such gatherings. The COVID-19 outbreak impact may cause severe long-term damages to the sports industry. As the virus has spread, an increasing number of matches and events have either been postponed or canceled. Some of the remarkable sports events are the Olympics, Formula One races, and the National Hockey League. The following figures (Figure 14, Figure 15 and Figure 16) manifest the impact of the COVID-19 outbreak on all these sports. 

As we discussed, many professional leagues across the globe suspended their seasons and events have been canceled. The Olympic Games that were initially supposed to take place in Tokyo at the end of July 2020 have been postponed by at least a year. Although there is no official decision regarding cancellation of the event, the event may be under serious threat due to the COVID-19 outbreak. If the event were to be canceled, the city of Tokyo stands to lose a potential 597 billion yen that it has invested into hosting the event [43].

Obviously, the potential loss of revenue from the combined hosting fees across the whole season could amount to over 602 million U.S. dollars. This amount is paid by the individual host nations [44]. 

Among several professional leagues across the globe that suspended their seasons, the National Hockey League in the United States is also believed to be affected due to the COVID-19 outbreak. It is estimated that for each home game that is canceled due to the pandemic, teams may lose an average 1.31 million U.S. dollars in ticket sales alone [45,46].

We discussed several industries that may suffer due to the COVID-19 outbreak. While the list does not end here, there are several other industries that may have been affected or may be impacted in future due to the outbreak. With globalization becoming a way of life, everything is interconnected and interdependent. Therefore, the impact on one industry may have an impact on several other industries to which it may be connected. 

## 4. Conclusions

In this article, we performed an extensive study on the COVID-19 outbreak. Based on the number of COVID-19 cases confirmed as well as recovered, and the ones that resulted in deaths, we have visually depicted how COVID-19, which was first identified in China, managed to spread across the globe rapidly. Our study suggested that there are at least 74.23% of active cases of the pandemic and the case fatality ratio is 5.36%. The world heat maps generated assert that several Asian countries and European countries have maximum cases of COVID-19. Our research not only focused on drawing observations from graph plots but also discussed the global impacts of the pandemic. The outbreak has not only claimed lives but has also affected several industries all over the globe. Several industries may suffer as a result of the COVID-19 outbreak, which may ultimately affect the global economy. Since the industries are interconnected and interdependent, it is necessary to understand the impact of a pandemic on one industry so as to anticipate which other industries may also be affected.

In the future, there is a need to perform a risk assessment for the COVID-19 outbreak. Several industries may suffer due to the pandemic, hence analyzing their previous data and current data, several observations can be made. This might lead to some level of risk assessment as well as risk management. Moreover, in the future, we would like to observe the trends of several other industries that may have been impacted by the outbreak such as the fashion industry, real estate, and the supply chain industry. 

## Figures and Tables

**Figure 1 healthcare-08-00148-f001:**
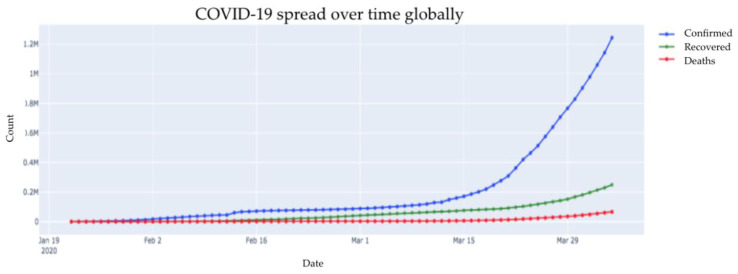
Outbreak of corona virus disease 2019 (COVID-19) across globe

**Figure 2 healthcare-08-00148-f002:**
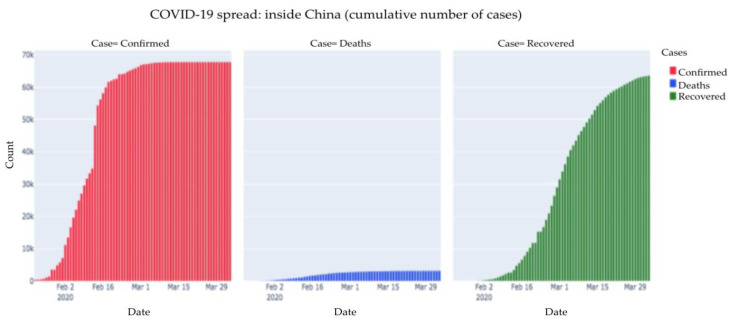
COVID-19 spread: inside China (cumulative number of cases)

**Figure 3 healthcare-08-00148-f003:**
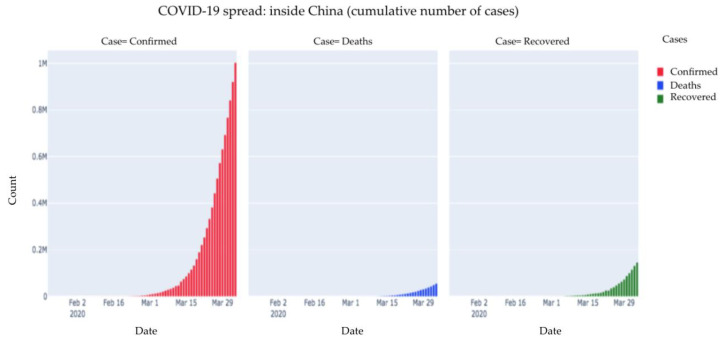
COVID-19 spread: outside China (cumulative number of cases).

**Figure 4 healthcare-08-00148-f004:**
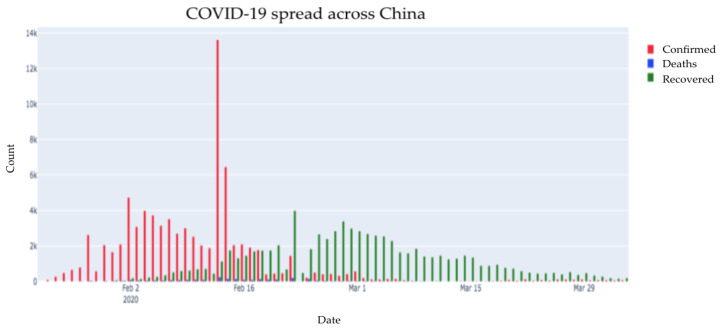
Distribution of all three cases for China.

**Figure 5 healthcare-08-00148-f005:**
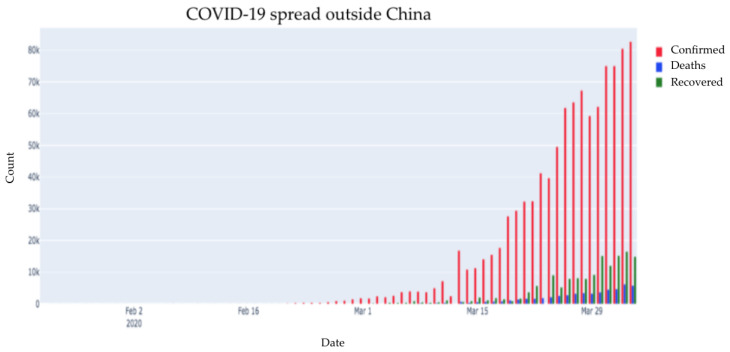
Distribution of all three cases outside China.

**Figure 6 healthcare-08-00148-f006:**
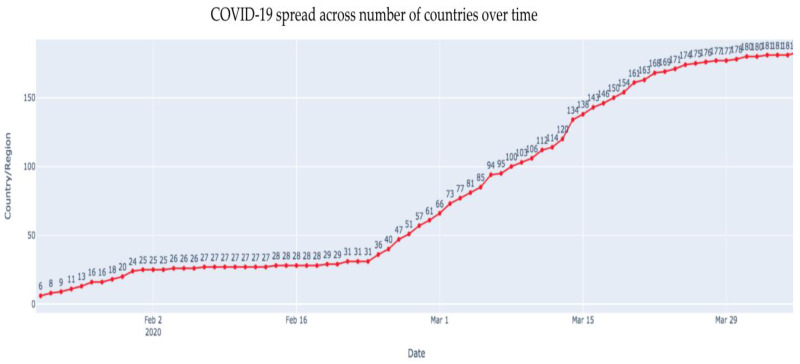
The number of countries affected across the globe.

**Figure 7 healthcare-08-00148-f007:**
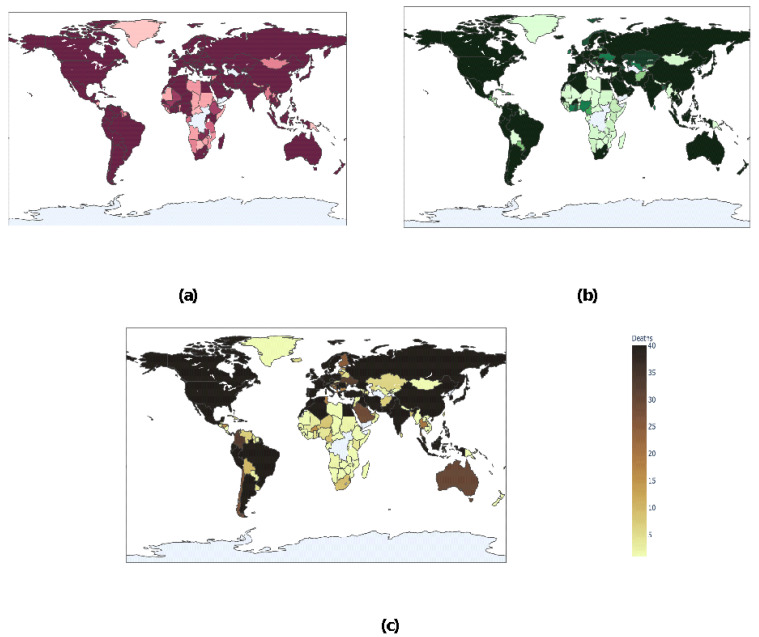
Heat Maps showing the number of (**a**) confirmed cases; (**b**) recovered cases; and (**c**) death cases across the world.

**Figure 8 healthcare-08-00148-f008:**
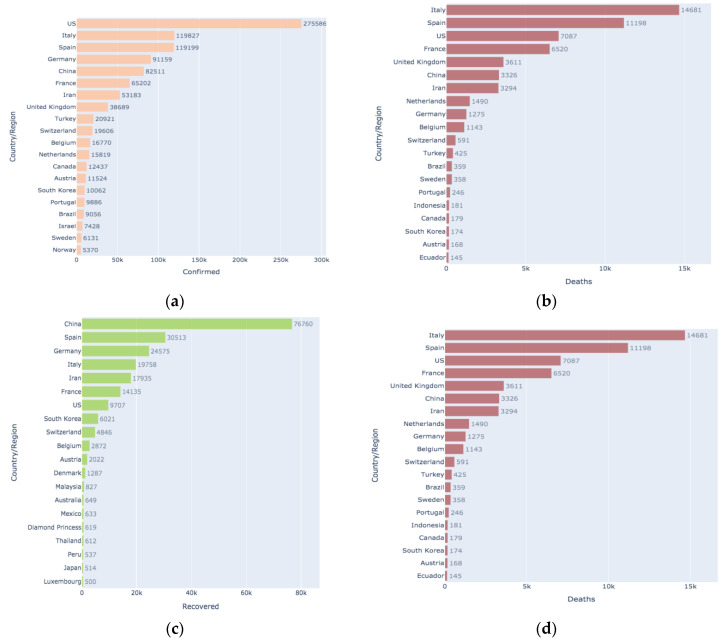
COVID-19 spread across the globe: (**a**) cases confirmed; (**b**) cases that resulted in death; (**c**) cases recovered; and (**d**) cases active globally.

**Figure 9 healthcare-08-00148-f009:**
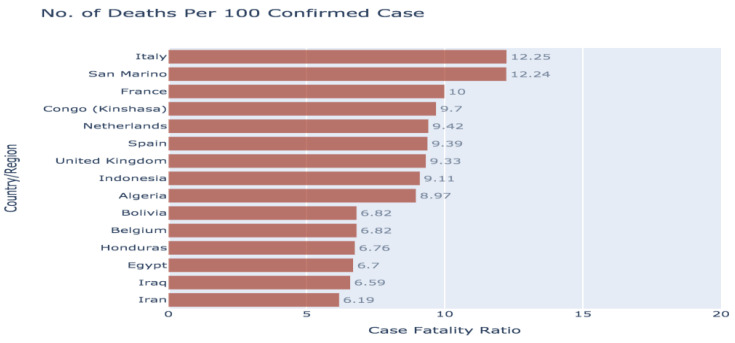
The number of deaths per 100 confirmed cases.

**Figure 10 healthcare-08-00148-f010:**
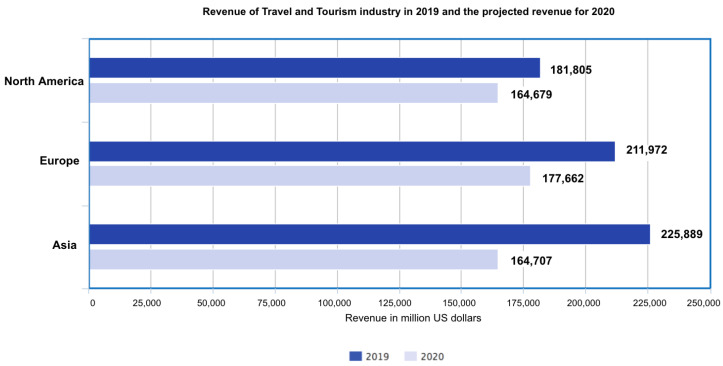
Comparison of global revenue for travel and tourism industry.

**Figure 11 healthcare-08-00148-f011:**
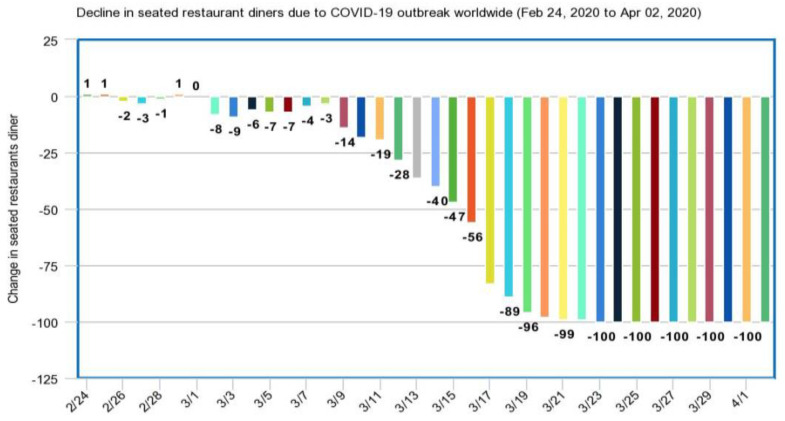
Decline in seated restaurant diners due to the COVID-19 outbreak worldwide.

**Figure 12 healthcare-08-00148-f012:**
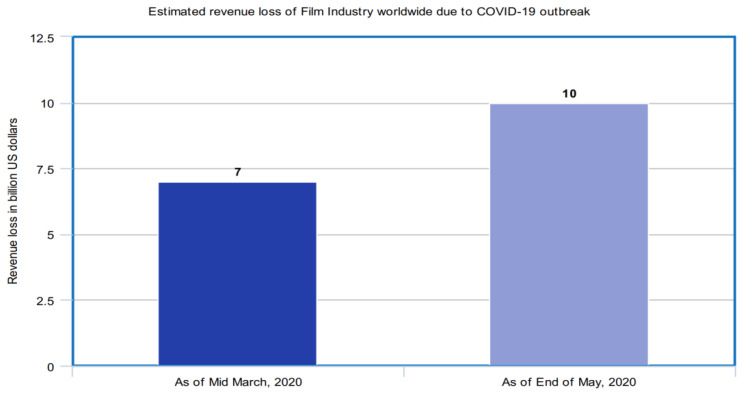
Estimated revenue loss of the film industry worldwide due to the COVID-19 outbreak.

**Figure 13 healthcare-08-00148-f013:**
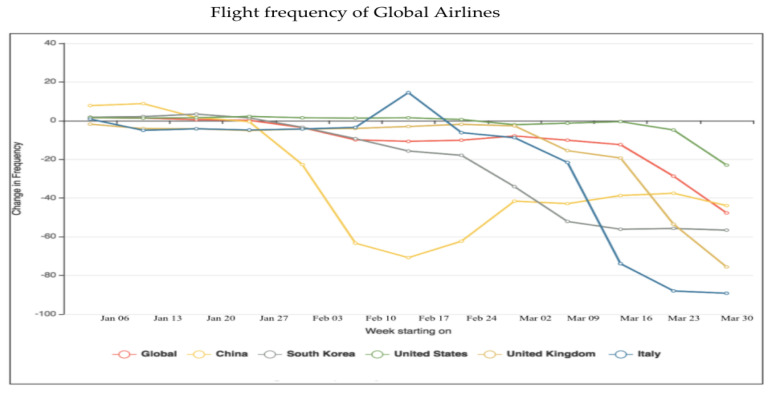
Change of flight frequency of global airlines due to the COVID-19 outbreak.

**Figure 14 healthcare-08-00148-f014:**
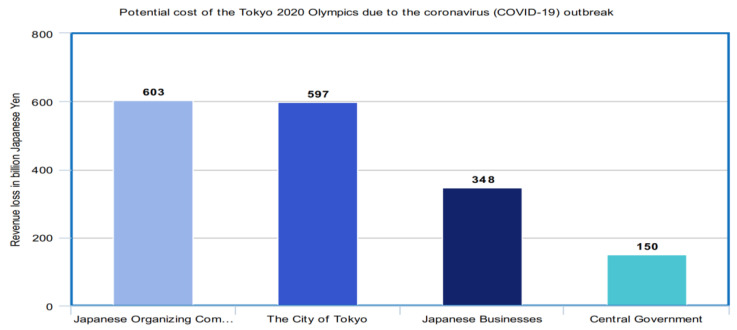
Potential cost of Tokyo 2020 Olympics due to the COVID-19 outbreak.

**Figure 15 healthcare-08-00148-f015:**
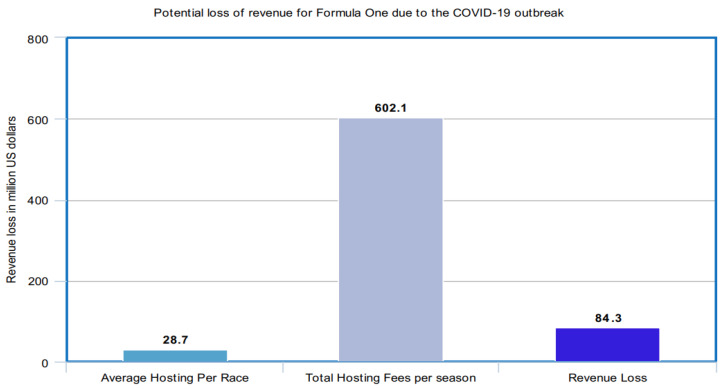
Potential loss of revenue for Formula One due to the COVID-19 outbreak.

**Figure 16 healthcare-08-00148-f016:**
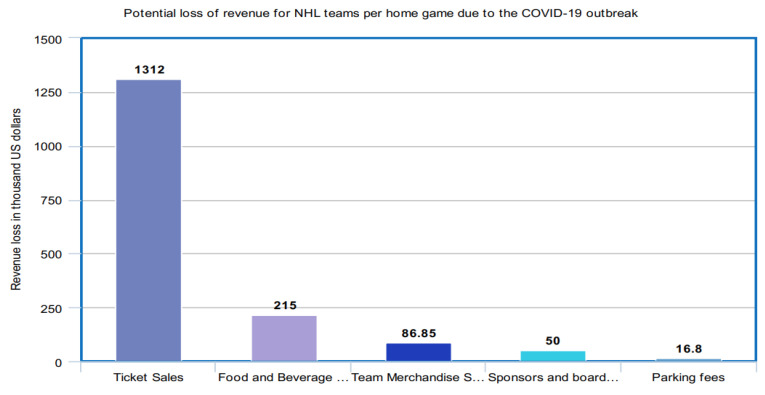
Potential loss of revenue for National Hockey League (NHL) teams per home game due to the COVID-19 outbreak.

**Table 1 healthcare-08-00148-t001:** Top 10 countries with active cases.

Country/ Region	Confirmed	Deaths	Recovered	Active
US	275,586	7087	9707	258,792
Italy	119,827	14,681	19,758	85,388
Spain	119,199	11,198	30,513	77,488
Germany	91,159	1275	24,575	65,309
China	82,511	3326	76,760	2425
France	65,202	6520	14,135	44,547
Iran	53,183	3294	17,935	31,954
UK	38,689	3611	208	34,870
Turkey	20,921	425	484	20,012
Switzerland	19,606	591	4846	14,169
Belgium	16,770	1143	2872	12,755

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
