# Peer review of "Analysis of Outbreak and Global Impacts of the COVID-19"

_healthcare, 2020, doi:10.3390/healthcare8020148_

Round 1
Reviewer 1 Report
This manuscript describes the spread of COVID-19 and how the COVID-19 outbreak impacts a variety of industries and global economy. The authors review a number of papers and give a detail summery regarding the global distribution of COVID-19 cases. However, the COVID-19 spreads rapidly and seams difficult to contain in some areas. Data presented in this manuscript appear to be out of date when published. For instance, current data indicate the most severely affected country by COVID-19 is USA while the data shown in this manuscript suggests US is only moderately affected by COVID-19. It is also not surprising that COVID-19 pandemic has caused severe negative impacts on global economy.
Minor points
P45 and p48 “Coronavirus(es) ” not “Corona virus”.
P108 and 110 typo errors; should be “COVID-19”.
Author Response
Thank you for the comment. COVID-19 outbreak initially suggested that the most affected country by COVID-19 was China. However, over the last few weeks, the dataset has changed remarkably. Keeping in mind the latest reports, the figures and tables have been updated and the US is shown to be one of the most affected countries by the outbreak. Since COVID-19 spreads rapidly, this data may change in the future too, however, the results presented are with respect to the latest dataset i.e. 22 January 2020 to 04 April 2020.
Minor points
- P45 and p48 “Coronavirus(es) ” not “Corona virus”.
Thank you for the comment. The modifications have been made accordingly.
- P108 and 110 typo errors; should be “COVID-19”.
Thank you for the comment. The modifications have been made accordingly.
Reviewer 2 Report
The importance of this study exists in section 4.5. Global Impacts of the COVID-19 outbreak. This point of view has a great importance in public health context.
However, there are many problems in this paper. Those should be appropriately addressed.
- For the section 4.5, the detailed information of data and method should be given. Part of data is referred in section 3.1, but not enough. What kind of projection was applied to make Fig.20 is not explained. Method is essential.
- L50, citing literature [2] is not adequate. Original research for SARS-CoV and MERS-CoV should be cited.
- Literature reviews are relatively old. For example, [16] is 2nd report from Imperial College among 13 reports to date. Why not citing others? Especially 12th report on 26 Mar shares the similar target with this paper.
- L96, three Coronaviruses should be three coronavirus diseases.
- L100 and other many places, "fatality rate" should be "case fatality ratio" or "case fatality risk". Nowadays, most theoretical epidemiologists uses the term of "case fatality ratio".
- L161, "the reproductive number" should be "the reproduction number". Nowadays, most theoretical epidemiologists prefer the latter.
- L173-179, why did the authors cite only one case report?
- L193-209, misunderstanding. Many studies apply WHO data or John Hopkins data. It's not a specific feature of this study.
- L203-205, already mentioned.
- L245, what is Statista? Citing the source information is needed.
- L247, for data sources, detailed information is needed.
- Many figures include essentially same information. It's too redundant. For example, Fig.1 and Fig.3 are not necessary. Fig.2 is enough.
- Fig.5-Fig.7 are vague. Explanation for scale is needed. Fig.7 has wrong colors. Are heat maps necessary? Fig.8-11 give similar information.
- L560, Tokyo Olympic is announced to be prolonged by 1 year.
- References are not well organized. Very incomplete.
Author Response
Comment 1: The importance of this study exists in section 4.5. Global Impacts of the COVID-19 outbreak. This point of view has a great importance in public health context.
However, there are many problems in this paper. Those should be appropriately addressed.
- For the section 4.5, the detailed information of data and method should be given. Part of data is referred in section 3.1, but not enough. What kind of projection was applied to make Fig.20 is not explained. Method is essential.
Thank you very much for the comment. The study is based on analyzing the outbreak of the novel coronavirus and its global impact. We have added a detailed description of the data and methods in section 4.5. All the figures have been explained for the study.
The coronavirus was detected on the 31st December 2019 in Wuhan City, Hubei Province, China. In the last couple of months, the number of cases has astonishingly increased to 4,000 cases in Wuhan City alone (uncertainty range: 1,000 – 9,700) [16]. The pandemic was not confined to Wuhan city of Hubei Province or China, but managed to spread globally, such that every continent (except Antarctica) has COVID-19 cases confirmed. The number of confirmed cases rose to 78000 [18] and now more than a million people have been affected by the outbreak. The COVID-19 outbreak not only resulted in more than three thousand deaths [19], but also affected several industries. Since the COVID-19 outbreak is highly attributed to infected passengers traveling internationally, the travel industry has been one of the most affected industries. This has led to travel restrictions for limiting the spread [36]. Multiple countries fear the risk of transportation [37] of infected patients and perform screenings at airports [38]. With globalization interacting and interconnecting people, businesses and nations, many industries become interconnected and interdependent. Therefore, an outbreak affecting a particular industry may also have a severe impact on other connected industries. In this section, we have considered some industries like the tourism industry, restaurants and leisure (industry), entertainment industry, travel industry, and the sports industry. The outbreak may have a significant effect on each of these industries. The study may also be useful in anticipating what all other industries that are connected to the industries taken in consideration may be impacted for future research. As mentioned in section 3.1, the data has been taken directly from Statista, an online portal for statistics that incorporates data collected by market and research institutes. Since for many industries we are comparing revenues, we have used bar charts (horizontal and vertical) to depict the same. The bar chart comparison based on statistics depicts the predicted revenue and the actual revenue for most industries. In this section, we discuss a few industries that were affected due to the deadly outbreak as follows:
Fig. 22 depicts three continents that have been affected the most due to the COVID-19 outbreak: North America, Europe, and China. As is evident from the graph above, there is a decline in the global revenue for the travel and tourism industry due to the COVID-19 outbreak. A bar graph has been used to show the comparison between the revenues for the years 2019 and 2020. We observe that the decline is 9.41 % percent for North America, for Europe, the decline is 16.18 %. Asia witnessed the maximum decline in revenue which is estimated to be 27.08% and therefore has suffered most in the case of the tourism industry. The estimates are based on IATA’s forecast regarding the overall effect of COVID 19 on the flight industry. As the demand for flights started decreasing, the number of leisure trips also declined to lead to a loss in revenue.
- L50, citing literature [2] is not adequate. Original research for SARS-CoV and MERS-CoV should be cited.
Thank you very much for the comment. We have added a few references to the literature. They are as follows:
- Chowell, G., Abdirizak, F., Lee, S., Lee, J., Jung, E., Nishiura, H., & Viboud, C. (2015). Transmission characteristics of MERS and SARS in the healthcare setting: a comparative study. BMC medicine, 13(1), 210.
- Millet, J. K., Goldstein, M. E., Labitt, R. N., Hsu, H. L., Daniel, S., & Whittaker, G. R. (2016). A camel-derived MERS-CoV with a variant spike protein cleavage site and distinct fusion activation properties. Emerging microbes & infections, 5(1), 1-9.
- Feng, Y., & Gao, G. F. (2007). Towards our understanding of SARS-CoV, an emerging and devastating but quickly conquered virus. Comparative immunology, microbiology and infectious diseases, 30(5-6), 309-327.
- Stadler, K., Masignani, V., Eickmann, M., Becker, S., Abrignani, S., Klenk, H. D., & Rappuoli, R. (2003). SARS—beginning to understand a new virus. Nature Reviews Microbiology, 1(3), 209-218.
- Raj, V. S., Osterhaus, A. D., Fouchier, R. A., & Haagmans, B. L. (2014). MERS: emergence of a novel human coronavirus. Current opinion in virology, 5, 58-62.
- Literature reviews are relatively old. For example, [16] is 2nd report from Imperial College among 13 reports to date. Why not citing others? Especially 12th report on 26 Mar shares the similar target with this paper.
Thank you very much for the comment. All the papers considered for the literature review have been published in the first quarter of 2020. Five more papers (and reports) including the 12th report on 26 March (Imperial College) have been appended to the literature survey. Further, we have cited some more recent papers in the introduction section as well as Section 4.5
- Ahmed, S. F., Quadeer, A. A., & McKay, M. R. (2020). Preliminary identification of potential vaccine targets for the COVID-19 coronavirus (SARS-CoV-2) based on SARS-CoV immunological studies. Viruses, 12(3), 254.
- Guarner, J. (2020). Three Emerging Coronaviruses in Two DecadesThe Story of SARS, MERS, and Now COVID-19. American Journal of Clinical Pathology.
- Zhang, J. J., Dong, X., Cao, Y. Y., Yuan, Y. D., Yang, Y. B., Yan, Y. Q., ... & Gao, Y. D. (2020). Clinical characteristics of 140 patients infected by SARS‐CoV‐2 in Wuhan, China. Allergy.
- Zu, Z. Y., Jiang, M. D., Xu, P. P., Chen, W., Ni, Q. Q., Lu, G. M., & Zhang, L. J. (2020). Coronavirus Disease 2019 (COVID-19): A Perspective from China. Radiology, 200490.
- Stoecklin, S. B., Rolland, P., Silue, Y., Mailles, A., Campese, C., Simondon, A., ... & Yamani, E. (2020). First cases of coronavirus disease 2019 (COVID-19) in France: surveillance, investigations and control measures, January 2020. Eurosurveillance, 25(6), 2000094.
- Liu, Y., Gayle, A. A., Wilder-Smith, A., & Rocklöv, J. (2020). The reproductive number of COVID-19 is higher compared to the SARS coronavirus. Journal of Travel Medicine.
- Smith, N. & Fraser, M., (2020), Straining the System: Novel Coronavirus (COVID-19) and Preparedness for Concomitant Disasters. Am J Public Health. Published online ahead of print February 13, 2020: e1–e2. doi:10.2105/AJPH.2020.305618.
- Zhang, S., Diao, M., Yu, W., Pei, L., Lin, Z., & Chen, D. (2020). Estimation of the reproductive number of Novel Coronavirus (COVID-19) and the probable outbreak size on the Diamond Princess cruise ship: A data-driven analysis. International Journal of Infectious Diseases.
- Huang, W. H., Teng, L. C., Yeh, T. K., Chen, Y. J., Lo, W. J., Wu, M. J., ... & Lin, C. F. (2020). 2019 novel coronavirus disease (COVID-19) in Taiwan: Reports of two cases from Wuhan, China. Journal of Microbiology, Immunology and Infection.
- Imai, N., Dorigatti, I., Cori, A., Donnelly, C., Riley, S., & Ferguson, N. M. (2020). Report 2: Estimating the potential total number of novel Coronavirus cases in Wuhan City, China. Imperial College London.
- Walker, P. G., Whittaker, C., Watson, O., Baguelin, M., Ainslie, K. E. C., Bhatia, S., ... & Cucunuba, Z. (2020). The Global Impact of COVID-19 and Strategies for Mitigation and Suppression. On behalf of the imperial college covid-19 response team, Imperial College of London.
- Bogoch, I. I., Watts, A., Thomas-Bachli, A., Huber, C., Kraemer, M. U., & Khan, K. (2020). Potential for global spread of a novel coronavirus from China. Journal of travel medicine, 27(2), taaa011.
- Gaythorpe, K. Imai, N., Cuomo-Dannenbur, G., Baguelin, M., BHatia, S., Boonyasiri, A., Cori, A., Cucunuba, Z., Dighe, A., Dorigatti, I., Fitzjohn, R., Fu, H., Green, W., Hamlet, A., Hinsley, W., Laydon, D., Gilani, G., Okell, L.Riley, S., Thompson, H., Elsland, S., Volz, E., Wan, H., Wang, Y., Whittaker, C., Xi, X., Donnely, C., Ghani, A., Ferguson, N . (2020). Report 8: Symptom progression of COVID-19
- Dorigatti, I., Okell, L., Cori, A., Imai, N., Baguelin, M., Bhatia, S., ... & Fu, H. (2020). Report 4: severity of 2019-novel coronavirus (nCoV). Imperial College London, London.
- Shi, H., Han, X., Jiang, N., Cao, Y., Alwalid, O., Gu, J., ... & Zheng, C. (2020). Radiological findings from 81 patients with COVID-19 pneumonia in Wuhan, China: a descriptive study. The Lancet Infectious Diseases.
The following research papers have been mentioned in the Introduction section
- World Health Organization. (2020). Coronavirus disease 2019 (COVID-19): situation report, 67.
- Hassan, S., Sheikh, F. N., Jamal, S., Ezeh, J. K., & Akhtar, A. (2020). Coronavirus (COVID-19): A Review of Clinical Features, Diagnosis, and Treatment. Cureus, 12(3).
- Tian, S., Hu, N., Lou, J., Chen, K., Kang, X., Xiang, Z., ... & Chen, G. (2020). Characteristics of COVID-19 infection in Beijing. Journal of Infection.
- MacIntyre, C. R. (2020). Global spread of COVID-19 and pandemic potential. Global Biosecurity, 1(3).
- MacIntyre, C. R. (2020). On a knife's edge of a COVID-19 pandemic: is containment still possible. Public Health Res Pract, 30(1), 3012000.
The following research papers have been appended to Section 4.5
- Chinazzi, M., Davis, J. T., Ajelli, M., Gioannini, C., Litvinova, M., Merler, S., ... & Viboud, C. (2020). The effect of travel restrictions on the spread of the 2019 novel coronavirus (COVID-19) outbreak. Science.
- Du, Z., Wang, L., Cauchemez, S., Xu, X., Wang, X., Cowling, B. J., & Meyers, L. A. (2020). Risk for transportation of 2019 novel coronavirus disease from Wuhan to other cities in China. Emerging Infectious Diseases, 26(5).
- Phelan, A. L., Katz, R., & Gostin, L. O. (2020). The novel coronavirus originating in Wuhan, China: challenges for global health governance. Jama, 323(8), 709-710.
- L96, three Coronaviruses should be three coronavirus diseases.
Thank you for the comment. The modification has been made accordingly.
- L100 and other many places, "fatality rate" should be "case fatality ratio" or "case fatality risk". Nowadays, most theoretical epidemiologists uses the term of "case fatality ratio".
Thank you for the comment. The modifications have been made accordingly.
- L161, "the reproductive number" should be "the reproduction number". Nowadays, most theoretical epidemiologists prefer the latter.
Thank you for the comment. The modifications have been made accordingly
- L173-179, why did the authors cite only one case report?
Thank you for the comment. The case reports are a part of the literature survey. the literature survey initially had two case reports [11] and [15], which highlighted COVID-19 cases in France and Taiwan. Few more reports [26] [28] [29] [30] have been appended to the study
- Stoecklin, S. B., Rolland, P., Silue, Y., Mailles, A., Campese, C., Simondon, A., ... & Yamani, E. (2020). First cases of coronavirus disease 2019 (COVID-19) in France: surveillance, investigations and control measures, January 2020. Eurosurveillance, 25(6), 2000094.
- Huang, W. H., Teng, L. C., Yeh, T. K., Chen, Y. J., Lo, W. J., Wu, M. J., ... & Lin, C. F. (2020). 2019 novel coronavirus disease (COVID-19) in Taiwan: Reports of two cases from Wuhan, China. Journal of Microbiology, Immunology and Infection.
- Walker, P. G., Whittaker, C., Watson, O., Baguelin, M., Ainslie, K. E. C., Bhatia, S., ... & Cucunuba, Z. (2020). The Global Impact of COVID-19 and Strategies for Mitigation and Suppression. On behalf of the imperial college covid-19 response team, Imperial College of London.
- Gaythorpe, K. Imai, N., Cuomo-Dannenbur, G., Baguelin, M., BHatia, S., Boonyasiri, A., Cori, A., Cucunuba, Z., Dighe, A., Dorigatti, I., Fitzjohn, R., Fu, H., Green, W., Hamlet, A., Hinsley, W., Laydon, D., Gilani, G., Okell, L.Riley, S., Thompson, H., Elsland, S., Volz, E., Wan, H., Wang, Y., Whittaker, C., Xi, X., Donnely, C., Ghani, A., Ferguson, N . (2020). Report 8: Symptom progression of COVID-19 , Imperial College of London.
- Dorigatti, I., Okell, L., Cori, A., Imai, N., Baguelin, M., Bhatia, S., ... & Fu, H. (2020). Report 4: severity of 2019-novel coronavirus (nCoV). Imperial College London, London.
- Shi, H., Han, X., Jiang, N., Cao, Y., Alwalid, O., Gu, J., ... & Zheng, C. (2020). Radiological findings from 81 patients with COVID-19 pneumonia in Wuhan, China: a descriptive study. The Lancet Infectious Diseases.
- L193-209, misunderstanding. Many studies apply WHO data or John Hopkins data. It's not a specific feature of this study.
Thank you very much for the comment. Mentioning the fact that the study applies WHO or John Hopkins data is purely to stress the reliability of data and is not intended as a feature of this study. Since many studies done in the past may not have reliable data to conduct their research, therefore the point was mentioned in the paper.
- L203-205, already mentioned.
Thank you for the comment. Though the paper initially discusses the aims, performing the literature survey made us realize the limitations of prior studies based on the critical analysis performed. Therefore, we state the limitations of the existing research works followed by the novelty of our study for each point to justify our work.
- L245, what is Statista? Citing the source information is needed.
Thank you for the comment. Statista is an online portal for Statistics that incorporates data collected by the market and other research institutes. It includes quantitative data related to media, business, finance, politics, and a wide variety of other areas of interest or markets. While the datasets have been taken from Statista, the data has been accumulated from reliable sources like World Health Organization (WHO), Opentable, The Hollywood Reporter, Official Airline Guide (OAG), Raidió Teilifís Éireann (RTE) Media services etc. We have cited the source information for all the graphs as needed [39] [40] [41] [42] [43] [44] [45].
- L247, for data sources, detailed information is needed.
Thank you very much for the comment. The data sources have been cited accordingly.
- Many figures include essentially same information. It's too redundant. For example, Fig.1 and Fig.3 are not necessary. 2 is enough.
Thank you very much for the comment. We have removed Fig. 2 from the manuscript.
- 5-Fig.7 are vague. Explanation for scale is needed. Fig.7 has wrong colors. Are heat maps necessary? Fig.8-11 give similar information.
Thank you very much for the comment. The explanation for the scale has been included. Fig. 7 has been modified. The heat maps have been included only to give a data visualization on a global scale, which may otherwise be neglected by other means like bar chart representations or treemaps. We agree that Fig 8-11 gives similar information, however, it is easier to depict the intensity in numbers when bar chart values are compared against one another.
- L560, Tokyo Olympic is announced to be prolonged by 1 year.
Thank you very much for the comment. We have modified the content accordingly based on the latest reports.
The Olympics Games that was initially supposed to take place in Tokyo at the end of July 2020 has been postponed by at least a year.
- References are not well organized. Very incomplete.
Thank you very much for the comment. The references have been organized accordingly.
Reviewer 3 Report
This article presents the number of COVID-19 cases of confirmed, recovered, death not only in China but also outside of China. The heat maps show the level of spread of this infectious disease by country/region. Also, the change of economic loss during the epidemic was described. The economic impact of this disease and countermeasures were very important indicator to fight against this disease for a certain, might be a long period.
However, in this study descriptive visualizations were conducted for the epidemic and economic data almost directly from the source. The novelty of this article is limited and the insights contained in this study are weak. The time-series data of the epidemic data from China presented was time series while that for other counties/regions was one-point data. In the middle of epidemic dynamics, it is difficult to capture the outbreak situation by one-point data and time-series data is preferable. In the case of heat maps, it could be depicted showing several time points. In some figures and tables, captions and legends are not well explained. They should be understandable by themselves without the main texts. The sections of the manuscript should follow the authors’ instructions.
Author Response
Comment 1: This article presents the number of COVID-19 cases of confirmed, recovered, death not only in China but also outside of China. The heat maps show the level of spread of this infectious disease by country/region. Also, the change of economic loss during the epidemic was described. The economic impact of this disease and countermeasures were very important indicator to fight against this disease for a certain, might be a long period.
- However, in this study descriptive visualizations were conducted for the epidemic and economic data almost directly from the source.
Thank you very much for the comment. All the descriptive visualizations conducted for the epidemic and economic data are based on the datasets available from the online portal for statistics called Statista. This is one of the most reliable sites that have available data to depict how different industries have been affected.
- The novelty of this article is limited and the insights contained in this study are weak.
Thank you very much for the comment. The study provides an overall analysis of the 2019 novel coronavirus and some of the associated global impacts. While some papers from the past do stress on the global impact of the outbreak, their study is limited and confined to specific domains. Our study, however, takes into account a number of industries that have been affected by the outbreak. To the best of our knowledge, this is the first paper that highlights several industries taken together like Travel and Tourism, Restaurants and Leisure, Entertainment Industry, Sports Industry that pinpoint at the large scale damage that a single outbreak is capable of doing. In order to justify the research work, we have appended more content to Section 4.5 and have updated the figures based on the most recent dataset.
- The time-series data of the epidemic data from China presented was time series while that for other counties/regions was one-point data.
Thank you very much for the comment. Most of the figures (Fig:1,3,12, 13,14,15,16,17,18,19,20,21) depicted in the study are time-series data. The reason behind presenting epidemic data from China is because the novel coronavirus was first discovered in China. In this study, we analyze the outbreak across the globe, hence cases from China are compared with the cases across the globe.
- In the middle of epidemic dynamics, it is difficult to capture the outbreak situation by one-point data and time-series data is preferable.
Thank you very much for the comment. Most of the figures have been updated based on the most recent data. Also, Figures 1,3,12, 13,14,15,16,17,18,19, 20,21 are indicative of time series analysis.
- In the case of heat maps, it could be depicted showing several time points.
Thank you very much for the comment. The heat maps have been updated and explained as per the latest datasets. Further, calendar heat maps have been added to the study.
- In some figures and tables, captions and legends are not well explained.
Thank you very much for the comment. The figures and tables have been updated accordingly.
- They should be understandable by themselves without the main texts.
Thank you very much for the comment. The figures and tables have been updated accordingly.
- The sections of the manuscript should follow the authors’ instructions.
Thank you very much for the comment. The manuscript has been drafted as per the authors’ instructions.
Round 2
Reviewer 1 Report
This manuscript has been significantly improved following a major revision.
Author Response
Comment 1: This manuscript has been significantly improved following a major revision.
Thank you for your support. We appreciate your valuable comments and suggestions that improved the quality of the manuscript.
Reviewer 2 Report
Largely improved.
However, several problems remain.
- The result CFR 5.36% may be from the data of Table 2, but the calculation (58787/1095915 = 0.0536) is not given there. It should be given there.
- I cannot understand the meaning of calendar heat maps.
- L507-508 and L511-512 can be integrated.
- L653 "case fatality ratio is 3.8%" does not match with the result.
In addition, the nature of Statista database is still vague. I think the detailed process of inquiry in Statista should be given, at least as appendix or supplementary material.
Author Response
Comment 1: However, several problems remain.
- The result CFR 5.36% may be from the data of Table 2, but the calculation (58787/1095915 = 0.0536) is not given there. It should be given there.
Thank you for the comment. We have modified the sentence to make it easier for comprehension. Please check page 11 as follows.
According to [18], the case fatality ratio of COVID-19 is 2%-3%. According to our analysis based on recent data, the number of cases where patients died with respect to the number of cases confirmed is found out to be 58787/1095915 = 0.0536, which is 5.36 %. The variation depicts how the pandemic outbreak has affected a large number of people in a short period of time.
- I cannot understand the meaning of calendar heat maps.
Thank you very much for the comment. Figures 18-21 have been removed from the manuscript to avoid duplicate figures with same meaning.
- L507-508 and L511-512 can be integrated.
Thank you very much for the comment. The lines have been integrated on page 11.
According to [18], the case fatality ratio of COVID-19 is 2%-3%. According to our analysis based on recent data, the number of cases where patients died with respect to the number of cases confirmed is found out to be 58787/1095915 = 0.0536, which is 5.36 %. The variation depicts how the pandemic outbreak has affected a large number of people in a short period of time.
- L653 "case fatality ratio is 3.8%" does not match with the result.
Thank you very much for the comment. The line has been modified to match the result in the Conclusion on page 17 as follows:
Our study suggested that there are at least 74.23% of active cases of the pandemic and the case fatality ratio is 5.36 %.
- In addition, the nature of the Statista database is still vague. I think the detailed process of inquiry in Statista should be given, at least as appendix or supplementary material.
Thank you very much for the comment. While sections 2.1 and 3.3 include information about Statista, specific links have been provided in the References section. A detailed description of Statista has been added to section 2.1 on page 4 along with supplementary material included along with the manuscript as follows:
- The revenue figures across the globe for the travel and tourism industry were taken from cruises, hotels, package holidays, and vacation rentals. The estimations were based on IATA’s (International Air Transport Association ) forecast on the flight industry.
- The decline in the number of seated diners is based on a survey performed by OpenTable (online restaurant-reservation service company) and the data pertains to online reservations, phone reservations and walk-ins. OpenTable is known to be active in more than 80 countries. Australia, Canada, Germany, Ireland, Italy, Japan, Mexico, the Netherlands, Spain, the United Kingdom and the United States are some of the countries where OpenTable is active.
- The estimated revenue loss of the film industry worldwide is based on a survey performed by The Hollywood Reporter (American digital and print magazine, and website).
- The flight frequency of global airlines is based on a survey performed by Official Airline Guide (OAG) schedules analyzer. OAG is the world's leading provider of digital flight information, intelligence and analytics for airports and airlines.
- The potential cost of Tokyo 2020 Olympics is based on a survey performed by RTE (Raidió Teilifís Éireann) which is a national public service media of Ireland.
- The potential loss of revenue for Formula One is based on a survey performed by Forbes which is a leading global media company.
- The estimated loss of revenue for National Hockey League (NHL) teams is based on a survey performed by The Athletic which is a sports journalism website.
Reviewer 3 Report
Comment to the authors:
According to the instructions for authors, article type research manuscripts should follow the section structure: Introduction, Materials and Methods, Results, and Discussion. This research manuscript does not follow the structure; thus, it is difficult to capture the objectives of this study and the study flow. Also, there are too many figures of which some are not necessary to show and make readers confused. To report and emphasize important points, it should be explained simply avoiding unnecessary explanation.
The literature review section is too long describing findings from previous studies that are not directly related to this study. This information should be summarized much compactly and combined into the introduction section.
Materials and method section includes a lot of data relating to COVID-19 (section 3.2.1–17) and they should be separated into the result section.
Tables 1-4 are not necessary as already explained in the text.
Figure 2 is not informative because the visual information contained in this figure is capturable from Figure 1. It can be removed.
Section 3.2.4–6 can be one section. Figures 4–6 can be one figure with 3 panels.
The figures were explained in the main text but it should be also explained in figure captions for readers who check the figures without the main text.
Section 3.2.7–10 can be one section. Figure 7–10 can be one figure (or one figure with 3 panels, for example).
Section 3.2.12, 13, 16, 17 are time-series cumulative/daily incidence inside/outside China. Since they are time-series and the same type as Figure 1, they can be placed just after section 3.2.1.
Section 3.2.14 and 15 can be combined with section 3.2.12. Figures 14 and 15 can be removed since they are duplicate from Figure 13.
Section 3.2.18–21 and figure 18–21 are duplicate from Figure 1 and Figure 3 (although Figure 3 shows cumulative number). If they do not have weekly trends, these are not meaningful and should be removed.
4.2. Comparison and Table 5 are not insightful because the scopes of the studies vary.
4.2.1–5 should be in the result section.
Author Response
Reviewer #3
Comment 1: According to the instructions for authors, article type research manuscripts should follow the section structure: Introduction, Materials and Methods, Results, and Discussion. This research manuscript does not follow the structure; thus, it is difficult to capture the objectives of this study and the study flow. Also, there are too many figures of which some are not necessary to show and make readers confused. To report and emphasize important points, it should be explained simply avoiding unnecessary explanation.
Thank you very much for the comment. The manuscript has been modified to follow the section structure respectively: Introduction, Materials and Methods, Results and Discussion, and Conclusion
Section 1 includes the Introduction and Literature Survey. Section 2 incorporates Materials and Methods for our proposed work while section 3, deals with Results and Discussions. In section 4, we present Conclusion.
- The literature review section is too long describing findings from previous studies that are not directly related to this study. This information should be summarized much compactly and combined into the introduction section.
Thank you very much for the comment. Since the study deals with the pandemic outbreak as well as its global effects, we have tried to incorporate recent studies that highlight either of the topics by introducing articles as well as case studies. The literature review has summarized compactly and combined into the introduction section. Please check the new Introduction on pages 2-3.
- Materials and method section includes a lot of data relating to COVID-19 (section 3.2.1–17) and they should be separated into the result section.
Thank you very much for the comment. The manuscript has been modified accordingly. Section 3 now incorporates Results and Discussions.
- Tables 1-4 are not necessary as already explained in the text.
Thank you very much for the comment. Tables 1, 2 and 4 have been removed from the manuscript as they are explained in the text. However, Table 3 provides more information than stated in the manuscript. The number of cases confirmed, recovered and deaths are much more conveniently explained in the table rather than it could be in the manuscript text.
- Figure 2 is not informative because the visual information contained in this figure is capturable from Figure 1. It can be removed.
Thank you very much for the comment. Figure 2 has been removed from the manuscript.
- Section 3.2.4–6 can be one section. Figures 4–6 can be one figure with 3 panels.
Thank you very much for the comment. Figures 4-6 have been modified accordingly.
- The figures were explained in the main text but it should be also explained in figure captions for readers who check the figures without the main text.
Thank you very much for the comment. The figures have been updated accordingly and now all the figures now have captions.
- Section 3.2.7–10 can be one section. Figure 7–10 can be one figure (or one figure with 3 panels, for example).
Thank you very much for the comment. Figures 7-10 have been modified accordingly.
- Section 3.2.12, 13, 16, 17 are time-series cumulative/daily incidence inside/outside China. Since they are time-series and the same type as Figure 1, they can be placed just after section 3.2.1.
Thank you very much for the comment. Figures 12, 13, 16 and 17 have been placed right after section 3.2.1.
- Section 3.2.14 and 15 can be combined with section 3.2.12. Figures 14 and 15 can be removed since they are duplicate from Figure 13.
Thank you very much for the comment. Figures 14 and 15 have been removed from the manuscript.
- Section 3.2.18–21 and figure 18–21 are duplicate from Figure 1 and Figure 3 (although Figure 3 shows cumulative number). If they do not have weekly trends, these are not meaningful and should be removed.
Thank you very much for the comment. Figures 18-21 have been removed.
- 2. Comparison and Table 5 are not insightful because the scopes of the studies vary.
Thank you very much for the comment. Keeping in mind that the study focuses on the analysis of the COVID-19 outbreak and its global impacts, the comparative analysis has been made with research articles that pertain to either of the cases. This also includes the updated Case Fatality Ratio i.e. 5.36%
To ensure that the scope of studies conform to our proposed work we have introduced two case studies to the table.
Walker, P. G., Whittaker, C., Watson, O., Baguelin, M., Ainslie, K. E. C., Bhatia, S., & Cucunuba, Z., The Global Impact of COVID-19 and Strategies for Mitigation and Suppression. On behalf of the imperial college covid-19 response team, Imperial College of London, 2020, 1-19 27.
Bogoch, I. I., Watts, A., Thomas-Bachli, A., Huber, C., Kraemer, M. U., & Khan, K., Potential for global spread of a novel coronavirus from China. Journal of travel medicine, 2020, 27(2), taaa011.
- 2.1–5 should be in the result section.
Thank you very much for the comment. The manuscript has been modified accordingly.
Round 3
Reviewer 2 Report
All points I suggested were properly addressed.
Only one thing, why L73-74 has so many capitalized words?
Author Response
Comment 1: All points I suggested were properly addressed. Only one thing, why L73-74 has so many capitalized words?
- Thank you for the comment. The lines have been modified accordingly as follows:
- The Preliminary Identification of Potential Vaccine Targets has been proposed for the novel coronavirus based on Severe acute respiratory syndrome-associated coronavirus (SARS-CoV) Immunological Studies [17]. Although their findings may help guide experimental efforts towards the development of vaccines against the novel coronavirus, theoretically, since the same has not been tested, there might be a question of feasibility for the research.
Reviewer 3 Report
This manuscript was improved. However, there are still some issues to be addressed.
Major comment:
The literature review in the introduction is still too long. This information should be given in order to explain why the current study is necessary (e.g., by describing what is studied so far and what is not known; thus, this study is necessary). Too detailed information from previous studies makes readers misfocus the scope of this study.
Also, relating to the suggestion above, section 3.4 (comparison analysis) should be removed.
Minor comment:
Some figures are low resolution and hard to read.
Figure 1 has no y-axis label. Should be “Count”.
Y-axis-label for Figures 2, 3 should be “Count”.
Section titles for 3.1.4 and 3.1.5 (also, figure legends/captions for Figure 4 and 5) could be changed as follows:
3.1.4. COVID-19 spread: inside China (cumulative number of cases)
3.1.5. COVID-19 spread: outside China (cumulative number of cases)
(Optional) The order of section 3.1.1–3.1.5 can be as follows: 3.1.1, 3.1.4, 3.1.5, 3.1.2, 3.1.3. This is because section 3.1.1, 3.1.4–5 are the cumulative numbers of cases all over the world/inside China/outside China, respectively, while section 3.1.2–3 are new cases (not the cumulative numbers).
In Figure 6, the numbers of countries are overlapped and cannot be read.
Figure 7: all panels could be of the same width.
Figure 8: overall figure legend is missing
Author Response
Comment 1: The literature review in the introduction is still too long. This information should be given in order to explain why the current study is necessary (e.g., by describing what is studied so far and what is not known; thus, this study is necessary). Too detailed information from previous studies makes readers misfocus the scope of this study.
Thank you very much for the comment, we have revised and shortened the Introduction part. Please check page 2.
Comment 2: Also, relating to the suggestion above, section 3.4 (comparison analysis) should be removed.
Thank you very much for the comment. Section 3.4 has been removed from the manuscript.
Comment 3: Some figures are low resolution and hard to read. Figure 1 has no y-axis label. Should be “Count”. Y-axis-label for Figures 2, 3 should be “Count”.
Thank you very much for the comment. The Y-axis labels have been added to the figures accordingly.
Comment 4: Section titles for 3.1.4 and 3.1.5 (also, figure legends/captions for Figure 4 and 5) could be changed as follows:
3.1.4. COVID-19 spread: inside China (cumulative number of cases)
3.1.5. COVID-19 spread: outside China (cumulative number of cases)
(Optional) The order of section 3.1.1–3.1.5 can be as follows: 3.1.1, 3.1.4, 3.1.5, 3.1.2, 3.1.3. This is because section 3.1.1, 3.1.4–5 are the cumulative numbers of cases all over the world/inside China/outside China, respectively, while section 3.1.2–3 are new cases (not the cumulative numbers).
Thank you very much for the comment. The modifications have been done accordingly.
Comment 5: In Figure 6, the numbers of countries are overlapped and cannot be read.
Thank you very much for the comment. A comprehensible version of the figure has been added to the manuscript.
Comment 6: Figure 7: all panels could be of the same width.
Thank you very much for the comment. The figures have been modified accordingly.
Comment 7: Figure 8: overall figure legend is missing
Thank you very much for the comment. The legend has been added to Figure 8 (COVID-19 spread across the globe: Cases Confirmed, Cases that resulted in Death, Cases Recovered, and Cases Active Globally).
Round 4
Reviewer 3 Report
The legend at the top of Figure 7 is not necessary as it already has the legend at the bottom.
The legend of Figure 8 should be placed below the figure (Please use the same style as Figure 7).
Figure 8. COVID-19 spread across the globe: Cases Confirmed, Cases that resulted in Death, Cases Recovered, and Cases Active Globally
Author Response
Comment 1: The legend at the top of Figure 7 is not necessary as it already has the legend at the bottom.
- Thank you for the comment. We have revised the legends in Figure 8 as requested on page 7.
Comment 2: The legend of Figure 8 should be placed below the figure (Please use the same style as Figure 7).
Figure 8. COVID-19 spread across the globe: Cases Confirmed, Cases that resulted in Death, Cases Recovered, and Cases Active Globally
- Thank you for the comment. We have revised the legends in Figure 8 as requested on page 8.
